# A BRIEF HISTORY OF THE SPECULATIVE MEASURES FOR AUTONOMY

## ABSTRACT

This paper presents a novel summary of the history of the evolution of the measures for autonomy (i.e. self-legislating systems), from Creation myths to the study of technological autonomy, progressing through five interrelated phases. First, the original legislator of the laws of nature is considered as the singular measure for autonomous existence. Second, a set of hierarchical governing systems are conceived as a sort of supplement to the limitations of the original monopolistic autonomy. Third, this hierarchy of governors is inverted by transcendental philosophy, putting emphasis on the immediate conscious "self" - *the auto* - in self-legislation. The fourth stage emerges in existential philosophy, which notices the seemingly inescapable paradoxes of this introverted autonomy, as all justice and justification becomes circularly self-justified. The fifth and most contemporary measure of autonomy universalizes this existential insight and measures each being's autonomy simply in proportion to its own alienated independence. The paper concludes with an analysis on the potential limits of this universalized autonomy and suggests a route for future research which may ultimately separate the measure of autonomy from the measure of responsibility for justice.

**I. Original Autonomy (OA):** Our search for the standard of autonomy (i.e. self-legislation) begins by speculating on its most pure and primordial state, namely, as a world legislating mode of existence. In the earliest known fragments of metaphysical prose, the writer Anaximander vividly characterizes the causal structure of the cosmos as a justice system,[1] and codifies the scientific analogy of the "laws of nature" — still used by scientists without much question to this day. Parallel notions of such original cosmic laws also developed eastward, in the literal meanings of the Dharma and Dao. In mytho-poetic accounts, it was not uncommon to find the highest divinity personified as the *Highest Judge* (as in the case of Díkē and Yahweh). Such an absolute self-legislator - the uncaused cause of all causes [2] - could only exist ontologically prior to, and outside of, all space, time, things, and laws, in total independence. But if OA is taken as the lone measure of autonomy and sole responsibility for all causes, then this would immediately render redundant the causal power of all other beings, reducing them to mere artificial effects of the original legislation.[3]

**II. Governmental Autonomy (GA):** The following measure emerges as a sort of narrative, conceptual, and practical supplement to OA's over-determinacy. This secondary measure breaks down the concept of autonomy into a relative independence claimed by a being's proximity to the OA itself. From this development flowed a conception of a hierarchy of discrete autonomies, i.e. from the divine, to the city-state, the ruler, the citizen, the slave, the animal, and so on. But a problem with this supplement was already realized by the first self-proclaimed philosophers, as they found that anyone could merely claim access to OA and so corrupt truth and justice in their own favor (for which the Sophists were infamous). A speculative solution was to determine autonomy relative only to the Ideals of, say, Truth and Justice in-and-of-themselves (auto kath' auto), and to do this through a strict rationality rather than blind acceptance of a hierarchy merely claiming their name. All in common, GAs took the increase or decrease of their contiguity with the highest forms and powers as the primary measure of discrete autonomy. But, consequently, discrete autonomies remained constrained to pre-ordained roles and purposes.

**III. Transcendental Autonomy (TA):** In a great reversal of these pre-established and external standards of hierarchy, the modern conception of autonomy begins a phase shift into what may be termed the *introverted polarity of autonomy*. Where we had previously measured with an eye toward a singular absolute autonomous justice over the rainbow, TA begins to transfer priority to the "self" in self-legislation. With the aid of the printing press, a living individual's knowledge or experience could take on a revolutionary impact rivaling that of the old holy texts, as we see in the examples of the encyclopedia and novel. There is a new bright boon in science, philosophy, and politics here, as the independent mind is unleashed and glorified to function as-if a God's eye view (i.e. the bedrock of certitude). From this view, two predominant tendencies of tracing autonomy emerge. First, in the mechanistic paradigm, OA+GA are given a new life, as all effects are traced back to ulterior mechanical laws, which, unlike the divinities of yesteryear, may now be empirically tested and verified by our own senses. Second, in the organicist paradigm, consciousness is taken as the pinnacle of organizational autonomy in nature, taking on a sort of equiprimordiality to OA itself. This is because living consciousness, capable of rationally discerning the limits of the laws of nature and culture that it is enmeshed within, thereby achieves a unique degree of independence from both. From this transcendent seat, consciousness is proven to be capable of some degree of self-legislation. But, if TA is correct, then autonomy finds itself trapped in a circle, whether by realizing its own redundancy as a byproduct of dead material forces or by realizing it is trapped within the circle of its own imaginations, as one's self becomes the sole beginning and end of justificatory processes.

**VI. Existential Autonomy (EA):** These limitations were clearly realized in the disenchanted epoch of existential and deconstructionist theory. While TA's mechanistic and organicist paradigms continue to hold onto their logical frameworks, the certitude and optimism which had colored the previous stage dims, after our inflated self-justifying concept of autonomy appears to lead us into deepened self-doubt, interpersonal alienation, and multiple World Wars. Reaching beyond the all-too-powerful measure of TA, EA weakens its concept of autonomy to include its inescapable and problematic nature, as we are not first freed by virtue of a superior rationality, but are rather *forced* to be free by merely being thrown into an introverted existence, while also anxiously subjected to the extroverted powers of others. In a word, we are forced to construct ourselves from a pure and empty freedom. At this stage, we look back with clear eyes through history, finding that our truths

---

[1] "Whence things have their origin, there they must also pass away according to necessity; for they must pay penalty and be judged for their injustice, according to the ordinance of time." trans. F. Nietzsche, 1996.

[2] The first mention of this definition of autonomy is in Aristotle's De anima 406a.

[3] Also of note is the problem of evil, of course, as the genealogy of responsibility for injustiuce traces back to this supposedly all-good and all-powerful entity. Satan may have tempted Adam, but who tempted Satan?

and values were not built on OA's own real justice-system, but on our own self-constructed myths. Knowing we ourselves crafted these ideas of autonomous justice, how could we possibly continue to believe that which we *know* we made up? We come to the chilling conclusion that there is no reliable way to ultimately measure autonomy whatsoever, as we cannot grasp even a single law of nature or culture all the way down to an absolutely just autonomous source.

**V. Universal Autonomy (UA):** In a radicalization of the insights of existential philosophy, the standard measure for autonomy continues to weaken, or, put positively, the measure for autonomy becomes more sensitive, granular, and all-encompassing. The core hypothesis of this present frontier of research is that each and every entity is thrown into its own independent existence, and so therefore autonomous to a minimal degree, in its own right.[4] These speculative measurements are achieved only by way of what are termed "flat ontologies",[5] where no being is initially discriminated as having more or less reality than any other. The two prevailing versions of these minimal standards of autonomy are the object-oriented[6] and ontogenetic variations.[7] The former measures the autonomy of an object in proportion to its own withheld superstability, i.e. insofar as the object is irreducible to its constituents or its relations. The latter determines the autonomy of a process in proportion to its suspended metastability, i.e. insofar as the process is open to both its constituents and relations. In other words, a being need not be total, superior, nor rational to have a measure of autonomy, but must simply be a singularity, merely opened to its own relations. Both variations implicitly refuse any global reduction to dead matter, laws, forms, etc., as each autonomy writes local laws upon others and carves up the world as we know it in accord with these countless jurisdictions. Here, any being can be found functioning as an agent, as a nexus for irreversible legislative interference with, or modulation of, its environment. Put more radically, UA may be interpreted as the philosophy of existence without any pre-ordained laws of nature at all, but rather only autonomous agents all the way up, down, through, and across the cosmos. Or, as the philosopher Gilbert Simondon summarizes, there is simply "no machine of all machines".[8]

**Autonomy is Not Enough:** If UA is the final measure of autonomy, and indeed the ontological status of each discrete being, then we run into some interesting metaphysical problems which rival those of OA in grandeur and perplexity, leading mutually exclusively to either a quantum standstill or an undifferentiated continuum.[9] The more pressing concern may be the complete untethering of autonomy from any ulterior standards and the consequent eradication of hierarchical connection to justice as such. The world is, in the simplest terms, a free-for-all. Indeed, we are even deprived of the resources to draw a practical distinction between the authentic and the artificial, the important and unimportant cause. A human mind, an ant, a fictional character, and an artificial intelligence all have a degree of equivalent autonomous independence from other beings, but the immediate practical concern here is the distribution of rights and responsibilities. Strangely, UA reveals in its minimalist measure the non-responsibility of autonomy in and of itself (i.e. its ontological irrelevance to our initiating paradigmatic of a justice-system). Not until UA have we had to reckon with a vast plurality of autonomies, regardless of potency or importance. Autonomy alone can no longer serve as the measure of responsibility, but only causality. If we take history as our guide, the next stage of thought may radicalize the following insight from UA into a model for autonomous systems: throughout our entire history up to this point, we have wrongly linked responsibility, virtue, rights, and justice itself with the maximal measure of autonomy. From here, we may begin to speculate on the conception of a measure of responsibility detached from, yet empowered by, all previous measures of autonomy. This axiom would entail more responsibility fall on some autonomies over others, even if these beings have measurably less freedom and control over probable outcomes. This independent measure of responsibility may be the *use* of autonomy to answer the cause of injustice. In this flat yet charged landscape of autonomous beings, a new order of rank might be established; a system where beings are no longer compelled as if by law to be the most powerful, intelligent, or free entity, but rather are measured in proportion to the simple sense of significance — perhaps, to legislate *justly;* to legislate a world of justice in complementarity with the vast totality of autonomies.

---

[4]This movement may be said to begin with A. Meinong's indexical ontology or A.F. Whitehead's process philosophy.

[5]A term coined by M. DeLanda, 2016.

[6]See: G. Harman, 2018

[7]See: G. Simondon, 2020 & Dunham, J; Grant, I.H.; Watson, S., 2011.

[8]Simondon, 2017, p.xvi

[9]See appendix UA.III.

URM STATEMENT

The authors acknowledge that at least one key author of this work meets the URM criteria of ICLR 2023 Tiny Papers Track.

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

# A    APPENDIX

GA – i. A narrative supplement to OA by GA can be found in the common eschatological account of the origin of injustice which also amplified human autonomy: a once boundless harmonious creation broke apart by some freely-chosen injustice, and so now creation strives for a sort of homecoming to that original state. We find such an account in Anaxagoras' philosophy as well as the Abrahamic myth of genesis, among others.

ii. Religious texts often doubled as recipes for discerning and obeying OA+GA's laws, in order to gain proximal autonomy oneself. This moral choice is obviously incoherent if OA is simply blamed for all outcomes.

iii. The first self-proclaimed philosophers noted the corruption of hierarchical GA and attempted a solution by following the priestess Diotima's dialectical process of prodding the intellect (lógos) beyond mere appearances, customs, and claims of justice toward Justice in-itself (auto kath' auto). The three famous breakthroughs here were devised by granting a rationalizable connection to OA through ceaseless intellectual conscience (Socrates), mathematization of Ideal forms (Plato), and understanding the teleology of substances (Aristotle).

UA – i. Analogical cognition appears to be held in common by UA theorists as superior and prior to logical cognition. The reason being that logic and physics function by reducing beings to structural laws rather than operative autonomies, whereas analogical cognition is capable of (speculatively) examining any thing as if a cause in its own right.

ii. In the wake of UA, we may now clarify the distinction between physics and metaphysics as the study of laws and autonomies, respectively.

iii. Addressing the causal problems of UA, it appears as though UA carries forward the existential in-distinction between the authentic and the artificial, and repeats the paradoxes embedded in initializing a causal sequence from pre-established laws (GA) or otherwise from self-circularity (TA). This is because ontogenetic relationality may be reduced to just another "law" of nature (as is the *continua* problem), and, conversely, object-oriented non-relationality finds beings incapable of direct relation without each immediately violating their own defining superstable autonomyies in principle (as is the *quanta* problem).

