# OpenReview forum: "A Brief History of the Speculative Measures for Autonomy"
_ICLR.cc/2023/TinyPapers — Submitted to Tiny Papers @ ICLR 2023_

### Official Review · Reviewer_iNCP · 2023-03-18

**Confidence:** 5

**Summary Of Contributions:**

philosophical paper on 5 measures of autonomy however no link to Machine learning

**Rating:**

Needs Clarification (NC): a submission which does not meet the reviewing criteria and needs clarification for its described problem or solution

**Strengths And Weaknesses:**

Paper discusses on the justice system with 5 measures of autonomy
However doesn't provide how these measures will link with Machine Learning
Paper is purely philosophical and NOT Recommended for ICLR Tiny Paper


**Suggested Changes:**

Authors can focus on using this measures of autonomy as policies that ascertain reinforcement learning

---

### Official Review · Reviewer_ZLu4 · 2023-03-28

**Confidence:** 3

**Summary Of Contributions:**

This paper summarizes the evolution of autonym’s measures throughout history. In addition, the author suggests a new perspective, in which autonym’s measures are not necessarily responsibility for justice’s measures.

**Rating:**

Great Start (GS): a submission which meets some of the reviewing criteria but has room for improvement

**Strengths And Weaknesses:**

Strengths:
1. Good flow, transition, and connections.
2. Good theoretical and law summary.

Weaknesses:
1. No emphasis on the impact for a specific domain, such as technology.
2. Interesting a view of point for autonym conceptualization, but more specification for this new direction and its impact would support the argument.

**Suggested Changes:**

Suggested Changes:

1. The paper provides a compressive study for autonym’s measures, but it does not render a real world application of autonym's measures and how people can utilize the measures to real world applications. The paper should emphasize the essential presence of autonym's measures, and link potential applications of autonym's measures in technology.
2. It is better for the paper to contain either simple empirical or user studies to validate that autonym’s measures are not necessarily responsibility for justice’s measures.

---

### Meta-Review · Area_Chair_eYt3 · 2023-04-06

**Recommendation:** Invite to revise
**Confidence:** 5

**Metareview:**

1. Paper provides information of relevant literature in field of Law but fails to provide in the field of artificial intelligence
2. paper does not discuss about novel findings in the field of artificial intelligence
3. assumptions stated are in field of law not Artificial Intelligence, no proofs provided on the assumptions
4. Author needs to work on making this paper relevant for ICLR conference, currently this paper is most suited for conference on Law
5. With limited time limit, revision for acceptance to CCR is hardly possible.

**Summary:**

paper on measures of autonomy, doesn't link these measures with technology or machine learning

**Comments And Feedback To The Authors:**

1. The paper is well written and well researched in the field of LAW, as it focuses on law with 5 measures of autonomy.
2. Current paper is well suited for a law conference but fails to add information on technology or link to artificial intelligence
3. The paper needs major revisions to include the impact these measures has on Artificial intelligence to real world applications
4. Focus on making this paper empirical findings with code & data, or theorical findings with proofs for the relevant assumptions made.

**Reason For Not Giving A Higher Recommendation:**

Agree with the rating and recommendations provided with both the reviewers.
This paper needs to be revised

**Reason For Not Giving A Lower Recommendation:**

N/A

---

### Decision · Program_Chairs · 2023-04-08

No revision received; not invited to archive